# Research on Erosion-Corrosion Rate of 304 Stainless Steel in Acidic Slurry via Experimental Design Method

**DOI:** 10.3390/ma12142330

**Published:** 2019-07-22

**Authors:** Ping Li, Yanjie Zhao, Libo Wang

**Affiliations:** 1Henan Key Laboratory of Materials on Deep-Earth Engineering, School of Materials Science and Engineering, Henan Polytechnic University, Jiaozuo 454000, China; 2State Key Laboratory of Solid Lubrication, Lanzhou Institute of Chemical Physics, Chinese Academy of Sciences, Lanzhou 730000, China; 3Henan International Joint Research Laboratory for High-Performance Light Metallic Materials and Numerical Simulations, Jiaozuo 454000, China

**Keywords:** 304 stainless steels (304 SS), erosion-corrosion, experimental design, quantitative analysis and qualitative analysis

## Abstract

A full three-factor two-level factorial experimental design method was carried out to investigate the effects of a single factor and their combining actions on weight loss of 304 stainless steels (304 SS). Quantitative analysis was performed to calculate the contribution values of temperature, rotation speed, sulfuric acid concentration, and synergistic actions. In particular, an 8 × 8 matrix was designed for the first time to define variation direction of erosion wear rate by qualitative analysis. The results show that slurry temperature has the most significant influence followed by sulfuric acid concentration. Rotation speed has the smallest effect. The synergies of the parameters all accelerate the weight loss rate, but they exhibit different intensity. This research provides useful guidelines for estimating the effects of environmental factors and material design in practical engineering application.

## 1. Introduction

Erosion-corrosion of slurry consisting of corrosive medium with solid particles is a common problem in many engineering fields such as metallurgy, power plant, mining, and chemical industry, and it is recognized as the major reason for the damage of the flow components, e.g., impeller of pump, pipeline, elbow, and choke (valve/nozzle) [1,2,3,4]. The common types of failure mechanism of erosion-corrosion wear involves electrochemical corrosion, mechanical wear, and the interaction or synergistic effect between them [5,6,7], the synergistic effect between erosion and corrosion, which is generally conformed to have a significant effect on erosion-corrosion wear properties of the material [8,9]. Although it has received more and more attention because of it in recent years [10,11,12,13], it is very difficult to predict erosion-corrosion rate and further in-service life of the materials due to the fact that the erosion-corrosion phenomenon is extremely complicated and affecting factors are changing with a variety of service conditions. Many investigations have focused on the effects of main factors, e.g., flow velocity/impact velocity of the solution [14], impact angle [4,15], particle size and/or particle concentration [16], pH [17], temperature of medium [3,18,19,20], and target material [21,22,23,24], etc.; however, only limited research was performed to study the individual effects of each factors and their interactions on total erosion–corrosion rate.

Mondal et al. [25] found that radial distance (RD), angle of inclination of the specimen (*θ*), and the distance traversed (DT) to be fatal in erosion and they can exhibit a certain change with a number of variables constant. Therefore, they used a full two-level factorial experimental design method to develop a linear regression equation for predicting wear rate of composite materials and to understand each of the individual variables, as well as their interactions on the wear rate by utilizing a rotating sample tester and selecting three factors. Subsequently, Mondal et al. found that RD is the most dominating factor to control the wear rate of composites, at the same time the interaction of these factors should also be considered. Similarly, Meng et al. [19] utilized a full two-level factorial experimental design method to assess erosion-corrosion performance of two stainless steels and analyze the individual effects of each parameter and their interactive contributions to the overall material degradation by means of impingement apparatus in 3.5% NaCl solution containing silica sand by selecting three environmental factors: Velocity, sand loading, and fluid temperature. After comparison, Meng et al. concluded that the total weight loss (*TWL*) and pure erosion loss (*E*) are significantly affected by sand loading and velocity. Furthermore, the interactive effect between them is the greatest, and among these three parameters fluid temperature has the smallest effect on *TWL* and *E*. Nevertheless, fluid temperature and its interactions with velocity or/and sand loading have non-negligible influence on pure corrosion loss (*c*) and synergy loss (*s*). Rajahram et al. [20] used a full factorial method with statistical analysis and interaction contour plots to explore the interaction among four main factors, i.e., velocity, sand size, sand concentration, and fluid temperature by using a slurry pot erosion tester where the samples were rotated by a motor and held at the end of the shaft. At the same time, the reasonable accuracy can be confirmed from the test results. A multiple liner regression equation without containing the interaction between the factors was obtained, in which it was indicated that velocity has the strongest influence on mass loss rate followed by sand concentration, temperature, and, finally, sand size. The interaction contour plots clearly show that the largest occurs between velocity and sand concentration for the largest gradient, the second is seen between sand concentration and sand size; subsequently, the third interaction is produced between temperature and sand concentration. However, unfortunately, the contour plots can only qualitatively explain rather than quantitatively analyze. From the previous literature, it was found that owing to the complexity of erosion wear environment, the results obtained by different researchers greatly depend on the selected factors, test device, and test method, etc.

Recently, Javiera and Magdalena [26] studied the effect of six factors, namely velocity, particle concentration, temperature, pH, concentration of dissolved oxygen (DO), and copper ion concentration on erosion corrosion (E-C) rate of API 5L X65 steel. The two effect graphs of a single factor and the interaction between two factors were observed to find how one or more factors influence the components. However, the interaction among three or more factors was not mentioned.

Atashin et al. [27] for the first time qualitatively and quantitatively investigated and surveyed the synergistic effects of parameters on the marine corrosion of AISI 316 stainless steel. They applied a full two-level factorial experimental design method and analysis of variance (ANOVA) to study the quantity of contribution in both individual and synergistic conditions. At the same time, a symmetric 16 × 16 matrix was designed to compare the difference values of corrosion rates of all arrays and predict effects of individual and combining action of factors on variation direction by qualitative analysis. However, to the best of our knowledge, few studies have been dedicated to both quantitative analysis on the contribution rate of individual and combining action of factors, and qualitative analysis on variation direction of erosion wear rate under liquid–solid slurry condition.

In this research, we also attempted to take the advantage of ANOVA. We designed a full two-level factorial experimental method to study the individual effects of three key environmental factors, viz., velocity, acid concentration, and temperature as well as their interactions on erosion wear rates of 304 stainless steel. Furthermore, an 8 × 8 matrix was designed and all arrays were calculated and compared to define variation direction of erosion wear rate. To date, we first performed both quantitative and qualitative methods to compare and confirm the significance of factors and their combining actions in the field of erosion-corrosion.

## 2. Experimental Method

### 2.1. Specimen Preparation

The typical austenitic 304 stainless steel material was used for this study. The stainless steel specimens were heated to 1050 °C and held for 1.5 h. It was then quenched into water. The hardness of the stainless steel was measured by Brinell tester five times and the average value was obtained to be 175 HB. The chemical compositions of the material are shown in Table 1.

### 2.2. Test Apparatus

Erosion-corrosion tests were carried out by a self-made slurry pot erosion apparatus (Model: MCF-14) and its schematic diagram is shown in Figure 1, which is similar to previous reference [28]. This apparatus mainly consisted of a speed regulating system, temperature controller, and slurry pot accessories. Motor with a maximum rotation speed of 1450 rpm drove the stirring impeller to mix slurry and keep the erodent suspended. This apparatus was equipped with a temperature controller at the top and pipes at the bottom of a water bank, which allows tap water to flow through the water regulating system for maintaining the fixed test temperature. The temperatures of the sample and the water were measured by thermocouples. Furthermore, the results were displayed and recorded in real time by the control cabinet with the computer. The specimen holder was fixed on the cover plate. The slurry was prepared according to the following requirement and poured into the slurry pot.

### 2.3. Experimental Methodology

A rectangular sample with dimensions of 10 mm × 10 mm × 20 mm was obtained by wire cutting from large sheets. The sample surfaces were polished with SiC papers up to 1200 grid and the entire sample rinsed with distilled water and ethanol, ultrasonically cleaned, dried, and weighed before and after the experiments to obtain the total weight loss by using a high precision digital balance (0.1 mg). Then, this rectangular sample was inlaid in the specimen holder, only a 10 mm × 10 mm surface of the sample was exposed in the slurry medium as a test surface and the rest of the surfaces were sealed with the specimen holder of Teflon. Therefore, the erosion wear surface area was 1 cm^2^. Every test’s duration was 2 h. The weight loss rate of the specimen was calculated as the ratio of erosion wear weight loss to the surface area of the specimen exposed in the slurry and test duration (in g⋅m^−2^⋅h^−1^). Each experiment was performed at least three times to ensure the experimental result repeatability.

This study mainly focused on three environmental parameters, i.e., rotational speed of motor (*S*), slurry temperature (*T*), and sulfuric acid concentration (*A*). In the present study, the impact velocity of slurry/particles is represented by different rotational speeds of motor [29]. Each parameter was set at two levels as shown in Table 2 with rotational speeds set at 800 rpm and 1200 rpm, slurry temperatures set at 25 ± 3 °C and 45 ± 3 °C, and sulfuric acid concentrations 0.25 mol/L and 0.50 mol/L. According to test design method [19], there were 2 × 2 × 2 = 8 trials for each set of experiments. The temperature of the test solutions was maintained at the set value throughout the entire duration of the experiment. The remaining components of the slurry were tap water and silica sand. The average size and mass fraction of the sands were 20~40 mesh and 10%. The experiment matrix is shown in Table 3. The weight loss rates under different conditions were received by erosion–corrosion tests and are listed as follows.

## 3. Results and Discussion

### 3.1. Weight Loss Rate

The weight loss rates (*W*_n_) under eight different experimental conditions are shown in Table 4. *W*_n_ is shown in Table 3 which represents the weight loss rates of trial condition n.

The results show that the weight loss rate of the material increases significantly with the increase of temperature and acid concentration. In addition, the rotational speed accelerates the weight loss rate. The open circuit potentials (OCP) tend to shift towards more negative values with the increase of temperature, the kinetics of reaction between the material and the electrolyte is favored with temperature [30,31,32]. The increase in temperature enhances the activity of the aggressive ions adsorbed on the material and accelerates the dissolution process as well as the breakout of passive film, as a result corrosion current densities, i.e., corrosion rate increases with temperature. When the rotational speed increases, the fluid kinetic energy increases and accelerates the removal of the surface passive film.

The increasing of rotation speed would replenish the H+ which was expended in the corrosion and increase the corrosion speed. It was observed that under the different conditions, the corrosion loss in 0.5 mol/L sulfuric acid solution was much more significant than that in 0.25 mol/L sulfuric acid solution due to the increased concentration.

### 3.2. Quantitative Analysis of Data

Analysis of variance (ANOVA) was carried out to study the effects of each test parameter/factor and their interactions on the response variable, i.e., weight loss rate, in order to find out the significance of each factor and their interactions on the erosion wear performance. First of all, we need to calculate the average impact of the factors for each level. For instance, the low level of the temperature factor is at the test conditions 1, 2, 5, 6, while the high level at 3, 4, 7, 8, as shown in Table 3. Therefore, the average effect of low and high levels for temperature can be calculated as follows:(1)$$W_{\mathrm{lT}}=\frac{1}{4}\left(W_{1}+W_{2}+W_{5}+W_{6}\right),$$
(2)$$W_{\mathrm{hT}}=\frac{1}{4}\left(W_{3}+W_{4}+W_{7}+W_{8}\right),$$
where *W*_1_, *W*_2_, *W*_5_, *W*_6_ are weight loss rates in low level corresponding to test conditions 1, 2, 5, 6, while *W*_3_, *W*_4_, *W*_7_, *W*_8_ are weight loss rates in high level corresponding to test conditions 3, 4, 7, 8 shown in Table 3, respectively. Similarly, the others including all interactions could be computed as *W*_lS_, *W*_hS_, *W*_lA_, *W*_hA_ in the same manner. Thus, the computed results are shown in Table 5.

Defining the sum of squares (SS) was the next step of the ANOVA method. The sum of squares could be obtained according to the following formula:*SS*_n_ = 2(*W*_ln_ − *W*_G_)^2^ + 2(*W*_hn_*− W*_G_)^2^(3)
where *W*_ln_ and *W*_hn_ are the average impact corresponding to the factor n viz. speed, temperature, and acid concentration, as well as their interactions for high and low levels and *W*_G_ is the average total weight loss rate for eight trial conditions, which is 67 g⋅m^−2^⋅h^−1^ according to Table 3. All the sums of square *SS*_n_ for each factor and their interactions are gathered in Table 6.

The final step of ANOVA was defined as the percentage contribution by using the following equation [27]:(4)$$K\left(\%\right)=\frac{SS_{K}}{SS_{s}+SS_{T}+SS_{A}+SS_{\mathrm{AT}}+SS_{\mathrm{SA}}+SS_{\mathrm{TA}}+SS_{\mathrm{STA}}}\times 100\%$$

*K* represents the factor whose effect can be calculated. Table 7 is the result for the percentage contributions of all factors and their interactions. From Table 7, we can easily find that the percentage contributions of four factors are higher than 20%, they are speed-temperature-acid concentration (*STA*, its *K* value reached 25%), temperature-acid concentration (*TA*, its *K* value reached 25%), speed-temperature (*ST*, its *K* value reached 20%), and temperature (*T*, its *K* value reached 20%). This indicated that the prominent contribution of the total erosion wear rate is due to temperature *T* and its interactions, i.e., *STA.*, *TA*, and *ST.* At the same time, contribution percentages of the interactions for the factors are higher than single-acting ones, and synergistic action is significant. It was followed by speed-acid concentration, acid concentration, and velocity, respectively. It is obvious that temperature has the largest contribution percentage of more than 20% in the range of three considered factors, i.e., temperature, speed, and acid concentration, and temperature plays a remarkable role on erosion wear rate variation of 304 stainless steel in this study. This is a good agreement with the experimental result conducted by Hu and Neville [33], etc. The second is acid, concentration with contribution percentage of 5%. The smallest is speed only contribution percentage of 0.4%, such trend could be due to so-called speed threshold effect [34,35,36,37,38,39]. Only when impact speed is higher than threshold value will it cause wear rate transition of the test material [34]. Although many investigations have indicated that impact velocity is a critical test variable on erosion wear rate [40], in this research the rotational speed of 1200 rpm, i.e., high level, could still not be high enough to reach the threshold velocity. Therefore, rotational speed has the smallest effect on erosion wear rate of 304 stainless steel in the range of considered factors in our research.

### 3.3. Qualitative Analysis of Data

Although quantitative analysis can provide statistical analysis of data [27] and objective and precise merits [41], it is difficult in description and justification of variation direction. Qualitative analysis is versatile/ubiquitous and easy to apply [42,43]. Moreover, it can predict variation direction of test result and obtain overwhelming parameter [27], as well as offer evaluative general information [44], suit most practical conditions, and simplify experimental and measurement procedures [45]. Whereas qualitative analysis is open to error and can be subjective [46]. Therefore, in general, two kinds of analysis approaches are combined to use [41,45].

In this study, as the first step for qualitative analysis, a symmetric 8 × 8 matrix was designed according to previous reference [27]. Each array represents the result that was obtained by comparing the erosion wear rates corresponding to the trial numbers shown in Table 3 and Table 4. For example, the array A_2,1_ represents (A > 0) the result comparing the erosion wear rate of trials 2 and 1. From Table 3, we can easily see that these two trials (2 and 1) only differed in acid concentration factor. Acid concentration factor of trial 2 has a high level while trial 1 has a low level. From Table 4, it can be seen that erosion wear rate of trial 2, i.e., *W*_2_, is larger than *W*_1_ of trial 1 (*W*_2_ > *W*_1_), this indicates that erosion wear rate increases with the rise of acid concentration value while the other factor values are fixed. Acid concentration factor has an accelerating effect on erosion wear rate. Similarly, the other arrays (A_3,1_, A_4,1_, A_5,1_, A_6,1_ A_7,1_, A_8,1_) of the first column can be obtained by comparing the corresponding erosion wear rate with that of the first trial. The comparison result is shown in Table 8. If the value of A_3,1_, and so on, is “> 0”, it shows having an accelerating effect on erosion wear rate, and if the value is “< 0”, it has an inhibitory effect. For example, the array A_6,1_ > 0 (*W*_6_ > *W*_1_) means SA > 0, which indicates that the synergistic action of speed and acid concentration has the accelerating effect on erosion wear rate. According to Table 8, all the arrays (A_2,1_, A_3,1_, A_4,1_, A_5,1_, A_6,1_ A_7,1_, A_8,1_) of the first column are greater than zero, and from Table 3 compared with trial 1, trials 2, 3, 5 have only the difference in high level of acid concentration, temperature, speed factor, respectively. This shows that regardless of acid concentration, speed, and temperature, every factor has an accelerating effect on erosion wear rate of 304 stainless steel (304 SS) materials. However, from the values of three arrays in the first column of Table 8 (A_2,1_, A_3,1_, and A_5,1_), we can clearly find that the rank of effect on erosion wear rate is temperature > acid concentration > speed. That is, among three factors, influence of temperature factor was the most significant followed by acid concentration, and with the above quantitative analysis this is in good agreement that speed has the smallest effect. It can be observed with other arrays in the first column, which correspond to the interaction factors, that A_4,1_, A_7,1_, and A_8,1_ of the combining action of temperature with other factors are much greater than the combining action value of acid concentration and speed factor without temperature factor, i.e., A_6,1._ This further illustrates that the effect of temperature factor is the strongest.

As for other arrays of matrix *A*_x,y_ (x represents rank while y represents column), the values of |*W*_x_ − *W*_1_| and |*W*_y_ − *W*_1_| need be calculated and compared, where *W*_x_, *W*_y_, and *W*_1_ are erosion wear rates of trial condition x, y, and 1, respectively, which are shown in Table 3 and Table 4. For example, *A*_3,2_ can be obtained by calculating |*W*_3_ − *W*_1_| and |*W*_2_ − *W*_1_|. The value |*W*_3_ − *W*_1_| represents *A*_3,1_, it is an array of the first column as mentioned above; the array shows the effect of temperature. While |*W*_2_ − *W*_1_| means *A*_2,1_, it is also an array of the first column, which indicates the effect of acid concentration. The result of |*W*_3_ − *W*_1_| > |*W*_2_ − *W*_1_| was obtained by calculating, and the conclusion of T > A can be inferred, i.e., temperature factor has a higher effect than acid concentration factor. Similarly, other arrays can be obtained as shown in Table 8. The greater difference value of |*W*_x_ − *W*_1_| and |*W*_y_ − *W*_1_|, the more significant the effect for the corresponding factor of trial condition x comparing with trial condition.

In Table 8, besides the arrays of the first column, for convenient comparison purposes, the significant comparisons of the other arrays are divided into three grades. The difference of two absolute value equal or less than 20 is considered to be insignificant (marked as ≈) with 20 to 50 generally significant (marked as >) and more than 50 be highly significant (marked as »). At the same time, increasing effect of combining action is shown as “↑”, decreasing effect of combining action as “↓”, and insignificant effect of combining action as “°”. Thus, obtained results are shown in Table 8. For three single factors corresponding to the arrays, i.e., A_3,2_, A_5,2_, and A_5,3_, according to the above method, e.g., A_3,2_, forA_5,2_,|*W*_5_ − *W*_1_| < |*W*_2_ − *W*_1_|, are easily drawn, since |*W*_5_ − *W*_1_| represents the effect of speed, |*W*_2_ − *W*_1_| represents the effect of acid concentration, and |*W*_5_ − *W*_1_| < |*W*_2_ − *W*_1_| the conclusion S < A can be drawn. Nevertheless, the difference value of |*W*_5_ − *W*_1_| and |*W*_2_ − *W*_1_| has only −4.35. Therefore, the effects of speed and acid concentration could be considered as S ≈ A, which have insignificant effect of speed compared with acid concentration. Finally, we can obtain these results: T > A, S ≈ A and S < T. As discussed above, temperature factor has the remarked effect on erosion wear rate.

To further analyze the synergistic effect of factors, three groups of arrays can be divided as follows [27].

The first group of arrays exhibits the increasing effect of factors combination compared with individual ones on erosion wear rate. For example, the array A_4,3_ presents the result of TA > T, which can be derived by means of the above mentioned method. To analyze A_4,3_, we need to study A_4,1_ and A_3,1_, respectively, that is compared with the two values |*W*_4_ − *W*_1_| and |*W*_3_ − *W*_1_|. As mentioned above, |*W*_4_ − *W*_1_| corresponds to the array A_4,1_ and shows the combining effect of temperature and acid concentration (*TA*). |*W*_3_ − *W*_1_| represents the arrayA_3,1_ and shows the effect of temperature (*T*). Due to |*W*_4_ − *W*_1_| > |*W*_3_ − *W*_1_|, the conclusion TA > T is obtained. This result shows the cooperation effect of temperature and acid concentration is greater than temperature alone on increasing the erosion wear rate. After adding acid concentration to temperature factor, it promotes the effect of temperature, i.e., TA > T. The arrays that belong to this group are shown by using the arrow pointing up in the cell. It should be pointed out that, for example, the array A_4,2_ shows the result of TA » A. This is since the difference values of |*W*_4_ − *W*_1_| and |*W*_2_ − *W*_1_| corresponding to the two arrays, i.e., A_4,1_ and A_2,1_, are larger than 50. It indicates that the cooperation effect of temperature and acid concentration is far greater than acid concentration alone on increasing the erosion wear rate. After adding temperature to acid concentration factor, it strongly promotes the effect of acid concentration, i.e., TA » A. Such situation also contains STA » A of A_8,2_, STA » T of A_8,3_, STA » S of A_8,5_, STA » SA of A_8,6_.

The second group of arrays displays the decreasing effect of factors combination in comparison with single ones on erosion wear rate. Such arrays are shown by using the arrow pointing down in the cell. In Table 8, these arrays can not be seen. All the effects of the combining actions of two and/or more than two factors are greater than one factor alone in a different degree.

The third group of arrays defines the insignificant effect of factors combination compared with individual ones on erosion wear rate. For example, A_6,2_ exhibits the effect of acid concentration nearly as much as combining action of acid concentration and speed (SA ≈ A). This shows to some extent (as mentioned above) nearly negligible effect of speed while acting with acid concentration. These arrays are marked with a circle sign in the cell.

The above analysis shows that three groups (there are not the second groups with decreasing effect) exhibit obvious differences in combining action of factors selected. Qualitative analysis can clearly display effect of factors and their combinations on variation direction of erosion wear rate.

## 4. Conclusions

The following conclusions are drawn from the research:Temperature has the largest contribution percentage of more than 20% in the range of three considered factors, i.e., temperature, rotational speed, and sulfuric acid concentration. At the same time, temperature plays a remarkable role on erosion wear rate variation of 304 stainless steel;According to the results of quantitative analysis, besides temperature, percentage contributions of three parameters (*STA, TA*, and *ST*) are higher than 20%, which are the interactions of temperature and other factors. Moreover, their values are higher than that of single temperature;All the three factors increase erosion wear rate of 304 SS (A > 0, T > 0, S > 0), namely from Table 3 where all three examined factors that show the accelerating effect on erosion wear under keeping two other factors at the fixed level;Under the range of our research, the contribution percentages of the interactions for two factors (*ST*, *SA*, *TA*) are higher than single-acting ones, the contribution percentage of the interaction for three factors (*STA*) are higher than that of two factors or single factor;The combining contributions of factors are larger than that of a single factor by quantitative analysis. Nevertheless, every factor exhibits different intensity on erosion wear rate by qualitative analysis.

## Figures and Tables

**Figure 1 materials-12-02330-f001:**
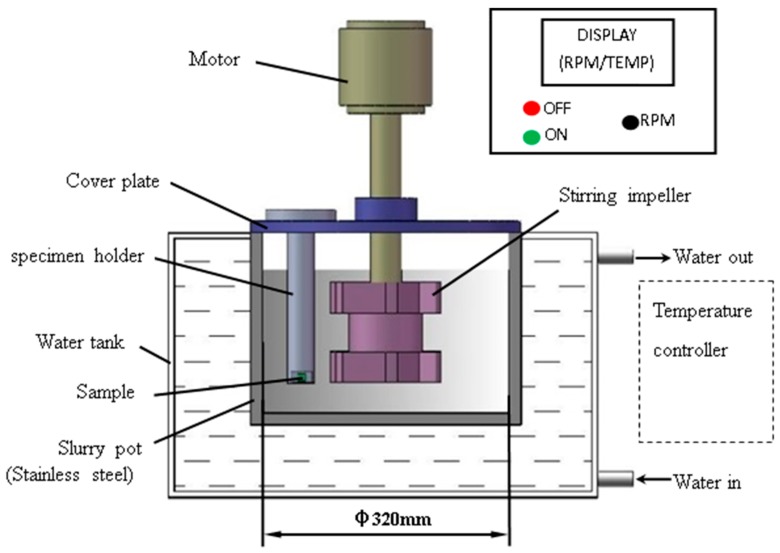
Schematic diagram of slurry pot erosion apparatus.

**Table 1 materials-12-02330-t001:** Compositions of the 304 stainless steel used (wt.%).

C	V	Cr	Mn	Ni	Cu	Fe
0.05	0.12	17.84	1.32	7.93	0.2	Bal.

**Table 2 materials-12-02330-t002:** Levels of test design parameters.

	Factors	Speed /*S* (rpm)	Temperature /*T* (°C)	Acid Concentration/*A* (mol/L)
Levels				
1		800	25 ± 3	0.25
2		1200	45 ± 3	0.50

**Table 3 materials-12-02330-t003:** Design matrix.

Test No	Variables			Outputs
	*S*	*T*	*A*	Weight Loss Rate
1	1 ∗	1	1	*W* _1_
2	1	1	2	*W* _2_
3	1	2	1	*W* _3_
4	1	2	2	*W* _4_
5	2 ∗	1	1	*W* _5_
6	2	1	2	*W* _6_
7	2	2	1	*W* _7_
8	2	2	2	*W* _8_

∗ The number 1 in this matrix represents low level, while 2 represents high level.

**Table 4 materials-12-02330-t004:** Loss rates under the eight conditions.

	Parameter	*W* _1_	*W* _2_	*W* _3_	*W* _4_	*W* _5_	*W* _6_	*W* _7_	*W* _8_
Weight Loss (mg)									
1		5.6	7.4	14.0	22.6	7.5	9.1	15.4	24.6
2		5.2	8.4	14.7	24.6	6.7	9.4	15.5	25.5
3		6.1	7.6	13.0	22.2	6.6	9.9	16.6	23.5
Average		5.6	7.8	13.9	23.1	6.9	9.5	15.8	24.5
Standard deviations		0.5	0.5	0.9	1.3	0.5	0.4	0.7	1.0
Weight loss rate (g⋅m^−2^⋅h^−1^)		28	39	70	116	35	47	79	123

**Table 5 materials-12-02330-t005:** Average effects of the factors for each level.

Factors	Calculating Formula	Calculating Value
*W* _lS_	$1/4\left(W_{1}+W_{2}+W_{3}+W_{4}\right)$	63
*W* _hS_	$1/4\left(W_{5}+W_{6}+W_{7}+W_{8}\right)$	71
*W* _lT_	$1/4\left(W_{1}+W_{2}+W_{5}+W_{6}\right)$	37
*W* _hT_	$1/4\left(W_{3}+W_{4}+W_{7}+W_{8}\right)$	97
*W* _lA_	$1/4\left(W_{1}+W_{3}+W_{5}+W_{7}\right)$	53
*W* _hA_	$1/4\left(W_{2}+W_{4}+W_{6}+W_{8}\right)$	81

**Table 6 materials-12-02330-t006:** Sum of squares for each factor and their interactions.

Parameters	Calculating Formula	Calculating Value
*SS* _S_	2(*W*_lS_ − *W*_G_)^2^ + 2(*W*_hS_ − *W*_G_)^2^	62
*SS* _T_	2(*W*_lT_ − *W*_G_)^2^ + 2(*W*_hT_ − *W*_G_)^2^	3538
*SS* _A_	2(*W*_lA_ − *W*_G_)^2^ + 2(*W*_hA_ − *W*_G_)^2^	802
*SS* _ST_	2(*W*_lS_ − *W*_G_)^2^ + 2(*W*_hS_ − *W*_G_)^2^ + 2(*W*_lT_ − *W*_G_)^2^ + 2(*W*_hT_ − *W*_G_)^2^	3600
*SS* _SA_	2(*W*_lS_ − *W*_G_)^2^ + 2(*W*_hS_ − *W*_G_)^2^ + 2(*W*_lA_ − *W*_G_)^2^ + 2(*W*_hA_ − *W*_G_)^2^	865
*SS* _TA_	2(*W*_lT_ − *W*_G_)^2^ + 2(*W*_hT_ − *W*_G_)^2^ + 2(*W*_lA_ − *W*_G_)^2^ + 2(*W*_hA_ − *W*_G_)^2^	4340
*SS* _STA_	2(*W*_lS_ − *W*_G_)^2^ + 2(*W*_hS_ − *W*_G_)^2^ + 2(*W*_lT_ − *W*_G_)^2^ + 2(*W*_hT_ − *W*_G_)^2^ + 2(*W*_lA_ − *W*_G_)^2^ + 2(*W*_hA_ − *W*_G_)^2^	4402

**Table 7 materials-12-02330-t007:** Each factor and their interactions.

Parameters	*S*	*T*	*A*	*ST*	*SA*	*TA*	*STA*
Contribution (*K*%)	0.4	20	4.6	20.4	4.9	24.7	25

**Table 8 materials-12-02330-t008:** Symmetric matrix of qualitative effect.

	1	2	3	4	5	6	7	8
1								
2	A > 0							
3	T > 0	T > A						
4	TA > 0	^↑^TA » A	^↑^TA > T					
5	S > 0	S ≈ A	S < T	S < TA				
6	SA > 0	°SA ≈ A	SA < T	SA « TA	°SA ≈ S			
7	ST > 0	ST > A	°ST ≈ T	ST < TA	^↑^ST > S	ST > SA		
8	STA > 0	^↑^STA » A	^↑^STA » T	°STA ≈ TA	^↑^STA » S	^↑^STA » SA	^↑^STA > ST	

^↑^ Increasing effect of combining action; ^↓^ decreasing effect of combining action; ° insignificant effect of combining action.

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
