# Peer review of "Research on Erosion-Corrosion Rate of 304 Stainless Steel in Acidic Slurry via Experimental Design Method"

_materials, 2019, doi:10.3390/ma12142330_

Reviewer 1 Report

Thank you for adding the data on reproducibility of the experiments. Although the paper is still based on a low number of experiments, it is now acceptable for publication in Materials.

Minor remarks follow:

Table 4: The average values should have the same number of valid digits as the measurements. Thus, for example for W1, the average should read 5.6, the standard deviation should be 0.5 and the weight loss rate 28.

Table 5: The precision is too high considering the error of measurement. Thus, W1S should read 63 etc.

Line 177: The same comment.

Line 186, 187, 195, 196: The same comment.

Table 6, Table 7: The same comment.

Author Response

Reply to the review report (1)

Question: Table 4: The average values should have the same number of valid digits as the measurements. Thus, for example for W1, the average should read 5.6, the standard deviation should be 0.5 and the weight loss rate 28.

Table 5: The precision is too high considering the error of measurement. Thus, W1S should read 63 etc.

Line 177: The same comment.

Line 186, 187, 195, 196: The same comment.

Table 6, Table 7: The same comment.

Answer: the modifications are marked in article

Reviewer 2 Report

The article is very interesting. I suggest the following changes:

Row 57

There is: et al and should be: et al. [20].

Row 72

The acronym DO should be explained.

Row 75

There is: weren’t and should be: were not.

Row 98

The measurement uncertainty of the hardness should be given.

Row 170

The method of calculating the factors for individual variables is described in detail in the text, while the description of factors from WlST to WhSTA by the statement “and so on in the same manner” may not be sufficient.

Row 171 Table 5

There is: average and should be: Average.

Row 286

There is: A8,2 and should be A6,2.

Author Response

Reply to the review report (2)

Question 1: Row 57 There is: et al and should be: et al. [20].

Answer: the modifications are marked in article

Question 2: Row 72 The acronym DO should be explained.

Answer: the modifications are marked in article

Question 3: Row 75 There is: weren’t and should be: were not.

Answer: the modifications are marked in article

Question 4: Row 98 The measurement uncertainty of the hardness should be given.

Answer: the hardness values for five times are 175,176177,174,173

Question 5: Row 170 The method of calculating the factors for individual variables is described in detail in the text, while the description of factors from WlST to WhSTA by the statement “and so on in the same manner” may not be sufficient.

Answer: deleting the description of factors from WlST to WhSTA in Table 5 and “and so on” by the statement “and so on in the same manner” ,because these has nothing to do with subsequent content and calculation of variance in table 6.

Question 6: Row 171 Table 5 There is: average and should be: Average.

Answer: the modifications are marked in article

Question 7: Row 286 There is: A8,2 and should be A6,2.

Answer: the modifications are marked in article

Reviewer 3 Report

The authors clarify both quantitative and qualitative methods to compare and confirm the significance of factors and their combining actions in erosion-corrosion effect of 304 stainless steel with good presented of theory experiment.

The work is well proceding for the methodology of theory experiment with different important factors influence on materials properties.

Author Response

Reply to the review report (3)

Comments and Suggestions for Authors

The authors clarify both quantitative and qualitative methods to compare and confirm the significance of factors and their combining actions in erosion-corrosion effect of 304 stainless steel with good presented of theory experiment.

The work is well proceding for the methodology of theory experiment with different important factors influence on materials properties.

Answer: no

This manuscript is a resubmission of an earlier submission. The following is a list of the peer review reports and author responses from that submission.

Round  1

Reviewer 1 Report

The manuscript describes results of eight 2-hour experiments on corrosion-erosion resistance of 304 stainless steel grade. The experiments were performed under a specific and rather narrow range of conditions of two levels of temperature, rotation speed rate (affecting particle velocity) and sulphuric acid concentration. Reasons for selecting the particular parameters are not given by the authors. Similarly, it is not explained what environment or application the tests should simulate. In consequence, the applicability of the data is very limited. None of the conclusions is generally valid, i.e. the effect of the studied parameters and their combinations may easily (and will) be different if any condition of the experimental system changes. In view of the very limited extent and depth of the research and applicability of the data, I propose to reject the manuscript. The scientific approach is correct and I would like to encourage the authors to continue working on the subject; however, it feels improper to publish such partial results, which can mislead readers.

Abstract, Line 122, conclusions: The authors refer to “acid concentration”. In view of corrosion stability of 304 stainless steel, it is of crucial importance what acid was used. This can be found only at a single location.

The error of measurement is not discussed. Mass loss data in Table 4 are given without standard deviations. It is not revealed if it was 1%, 10%, 50%... This is crucial for a reader to consider the weight of the data and all findings. Very probably, the number of digits given within the manuscript is completely inappropriate. Two valid digits for mass loss (Table 4), four-digit (!) average effects (Table 5), five-digit average mass loss (line 163), three-digit percentage contributions (line 172 and further), etc., are out of scope. Such precision is simply impossible in corrosion studies.

The statement “because with the rise of temperature, electrochemical reaction is intense and the passivation film of material surface is easy to be damaged” is oversimplifying.

Why columns and lines in Table 8 are numbered instead of using concerned parameters?

The conclusions are too long and relying too much on parameters introduced and explained within the body of the manuscript. Main findings should be pointed out there in as simple wording as possible. Point 6 does not belong to conclusions at all.

The English needs to be improved. It is understandable but there are many strange formulations and errors making the reading unpleasant. Some sentences/formulation are very hard to understand, e.g., “At the same time, from the test results and confirmed reasonable accuracy.”; “Brinell hardness tests were measured…”; “was obtained by wire cutting, which is inlaid in the specimen holder.”; “… low and high levels about temperature …”; “The calculated result of … can be drawn the conclusion of …”; “From Table 8 there exist not these arrays, …”.

Author Response

Reply to the review report (1)

Question 1: Abstract, Line 122, conclusions: The authors refer to “acid concentration”. In view of corrosion stability of 304 stainless steel, it is of crucial importance what acid was used. This can be found only at a single location.

Answer: the modifications are highlighted in article

Question 2:The error of measurement is not discussed. Mass loss data in Table 4 are given without standard deviations. It is not revealed if it was 1%, 10%, 50%... This is crucial for a reader to consider the weight of the data and all findings. Very probably, the number of digits given within the manuscript is completely inappropriate. Two valid digits for mass loss (Table 4), four-digit (!) average effects (Table 5), five-digit average mass loss (line 163), three-digit percentage contributions (line 172 and further), etc., are out of scope. Such precision is simply impossible in corrosion studies.

Answer: The standard deviation has been shown in Table 4, and the accuracy of the data has been modified and redlined in the article.

Question 3: The statement “because with the rise of temperature, electrochemical reaction is intense and the passivation film of material surface is easy to be damaged” is oversimplifying.

Answer: The supplementary contents are marked in article.

Question 4: Why columns and lines in Table 8 are numbered instead of using concerned parameters?

Answer: the numbers of columns and linesare matrix Numbers that need to be analyzed in conjunction with Tables 2 and Tables3

Question 5: The conclusions are too long and relying too much on parameters introduced and explained within the body of the manuscript. Main findings should be pointed out there in as simple wording as possible. Point 6 does not belong to conclusions at all.

Answer: the modifications are marked in article

Question 6: The English needs to be improved. It is understandable but there are many strange formulations and errors making the reading unpleasant. Some sentences/formulation are very hard to understand, e.g., “At the same time, from the test results and confirmed reasonable accuracy.”; “Brinell hardness tests were measured…”; “was obtained by wire cutting, which is inlaid in the specimen holder.”; “… low and high levels about temperature …”; “The calculated result of … can be drawn the conclusion of …”; “From Table 8 there exist not these arrays, …”.

Answer: the modifications are marked in article,e.g.,“At the same time,the reasonable accuracy can be confirmed from the test results.”“The hardness of the stainless steel was measured by Brinell tester for five times”“the average effect of low and high levels for temperature”“The result of |W3-W1|>|W2-W1| was obtained by calculating, and the conclusion of T>A can be inferred”“In Table8, these arrays can be not seen.”

Reviewer 2 Report

The authors report a full three-factor two-level factorial experimental design method to investigate the effect of a single factor and their combining actions on weight loss of 304SS. Quantitative
analysis was performed to determine the contribution values of each factor and the synergistic actions. The results showed that slurry temperature has the most significant influence followed by acid concentration, rotation speed has the smallest effect.

The manuscript is within the scope of Materials. However, before publication some minor concerns must be addressed:

The introduction needs to be updated. As presented does not reflect the actual state of the art.

The conclusions must be shortened.

Author Response

Reply to the review report (2)

Question 1:The introduction needs to be updated. As presented does not reflect the actual state of the art.

Answer: There are little articles which are studying the interaction of several factors, and we only find one article about the interaction of factors.

In recent year, Javiera and Magdalena [26] studied the effect ofsix factors namely velocity, particle concentration, temperature, pH, concentration of DO, and copper ion concentration on erosion corrosion (E-C)rate of API 5L X65 steel. The two effect graphs of single factor and the interaction between two factors were observed how one or more factors influence. However, the interaction among three or more factors weren’t mentioned.

26.Aguirre, J.;Walczak, M. Multifactorial study of erosion–corrosion wear of a X65 steel by slurry of simulated copper tailing[J]. Tribology International, 2018,126,177-185.

Question 2:The conclusions must be shortened.

Answer: the modifications are marked in article

Reviewer 3 Report

This paper studies the effect of a three factor, two level experiment (Temperature, Rotation speed, and acid concentration) on corrosion of SS304. Below is my comments for the authors:

1- the English language of the paper needs to be reviewed by a native speaker. Some sentences are very long and some choices of words need to be revised. 

Line 12 to 16: This part needs to be re-written. The sentence is way too long with many commas, make it shorter and use full stop instead of comma. Add more detail regarding the three factor you are studying.

Line 92: Table 1: please add the unit for the composition of SS304.

Line 141: when rotation speed increases....: Please also comment on the effect of rotation speed on diffusion of corrosive environment (fresh oxygen, H+) to the surface of the sample as well.

Line 148: For instance, the high level of the temperature factor is conditions 1,2,5,6: This needs to be changed to the low level of the temperature factor is conditions 1,2,5,6 based on the definition in table 3.

Author Response

Reply to the review report (3)

Question 1:Line 12 to 16: This part needs to be re-written. The sentence is way too long with many commas, make it shorter and use full stop instead of comma. Add more detail regarding the three factor you are studying.

Answer: A full three-factor two-level factorial experimental design method was carried out to investigate effect of single factor and their combining actions on weight loss of 304SS. Quantitative analysis was performed to calculate the contribution values of temperature, rotation speed, acid concentration and synergistic actions. In particular, an 8×8 matrix was for the first time designed to define variation direction of erosion wear rate by qualitative analysis.

Question 2:Table 1: please add the unit for the composition of SS304.

Answer:Table 1.compositions of the 304 stainless steel used(wt%).

Question 3: when rotation speed increases....: Please also comment on the effect of rotation speed on diffusion of corrosive environment (fresh oxygen, H+) to the surface of the sample as well.

Answer:The increasing of rotation speed would replenish the H+ which was expended in the corrosion and increase the corrosion speed.

Question 4: For instance, the high level of the temperature factor is conditions 1,2,5,6: This needs to be changed to the low level of the temperature factor is conditions 1,2,5,6 based on the definition in table 3

Answer: For instance, the low level of the temperature factor is at the test conditions 1,2,5,6, while thehigh levelat 3,4,7,8 shown in Table 3

Round  2

Reviewer 1 Report

In my review of the first version of the manuscript, I have expressed my opinion that publishing results of 8 simple, short-term experiments is not appropriate. I maintain this opinion. I also believe that none of the conclusions is generally valid, i.e. the effect of the studied parameters and their combinations may easily be different if any condition of the experimental system changes. The changes in the revision are minor, not addressing my main concerns.

In the review, I have noted that “the error of measurement is not discussed. Mass loss data in Table 4 are given without standard deviations. It is not revealed if it was 1%, 10%, 50%... This is crucial for a reader to consider the weight of the data and all findings. Very probably, the number of digits given within the manuscript is completely inappropriate. Two valid digits for mass loss (Table 4), … are out of scope. Such precision is simply impossible in corrosion studies.” The authors reacted by adding standard deviation of the averages for each experimental condition (!), see Table 4. It makes no sense and does not address my original remark. There is no information in the manuscript weather e.g. for W1 the mass loss reached 28.10 +- 2 (then, the number should read 28), or 28.10 +- 0.2 (it should read 28.1), or 28.10 +- 0.02 (then, it would be written correctly).

I am still convinced that the paper should be rejected in the current form.